# The Study of Physicochemical Properties and Blood Compatibility of Sodium Alginate-Based Materials via Tannic Acid Addition

**DOI:** 10.3390/ma14174905

**Published:** 2021-08-28

**Authors:** Beata Kaczmarek-Szczepańska, Adrianna Sosik, Anna Małkowska, Lidia Zasada, Marta Michalska-Sionkowska

**Affiliations:** 1Department of Biomaterials and Cosmetics Chemistry, Faculty of Chemistry, Nicolaus Copernicus University in Torun, 87-100 Torun, Poland; 291031@stud.umk.pl (A.S.); 291020@stud.umk.pl (A.M.); 296559@stud.umk.pl (L.Z.); 2Department of Environmental Microbiology and Biotechnology, Faculty of Biology and Veterinary Science, Nicolaus Copernicus University in Toruń, 87-100 Torun, Poland; mms@umk.pl

**Keywords:** thin films, sodium alginate, tannic acid, blood compatibility

## Abstract

In this study, sodium alginate-based thin films were modified by the addition of tannic acid. Materials were obtained by solvent evaporation. They were characterized by the observation of its morphology and its surface by scanning electron microscope and atomic force microscope. The thermal properties were studied by differential scanning calorimetry. The concentration of tannic acid released from the material was determined by the Folin–Ciocalteu method. The material safety for biomedical application was determined by the hemolysis rate study in contact with sheep blood as well as platelet adhesion to the material surface. Based on the obtained results, we assume that proposed films based on sodium alginate/tannic acid are safe and may potentially find application in medicine.

## 1. Introduction

Synthetic products, including polymers, vary in their fate and their repetitive properties. Nowadays, because the tread with fossil fuel is maintained, their overall use worldwide has a negative impact on the environment [1]. Trends take into account the environment derived from natural sources and what benefits will be derived in the coming years [2].

Polysaccharides are polymeric carbohydrate structures of simple sugars (monosaccharides) linked together by glycosidic bonds. They are the most common natural product produced by plants and fungi [3]. Based on the structural composition, we can distinguish homoglycans—polysaccharides made of one type of monomer—and heteroglycans made of several types of monomers [4].

The group of polysaccharides is characterized by such properties as non-toxicity, biocompatibility, biodegradability, multifunctionality, high chemical reactivity, chirality, chelating, and adsorption capacity [5,6,7]. Polysaccharides are widely used in the preparation of hydrogels due to the presence of hydrophilic functional groups that are able to absorb water and can be easily cross-linked by various chemical and physical methods. Moreover, they are obtained from widely available and renewable sources [8].

Alginate is a naturally occurring anionic, hydrophilic polysaccharide. The structure of alginate consists of blocks of β-D-mannuronic acid monomer (M) linked by bonds (1–4) and α-l-guluronic acid (G), which can be arranged as homopolymer sequences (MMM or GGG) or alternating sequences (MGMG) along the polymer chain. It is mainly obtained from brown algae (*Phaeophyceae*). Alginates can also be obtained by bacterial biosynthesis from *Azotobacter* and *Pseudomonas*, showing a more defined chemical structure and physical properties compared to alginates obtained from seaweed [9,10]. Due to its unique properties such as biocompatibility, biodegradability, non-antigenicity, chelating capacity, non-toxicity, and gelling properties, alginate is widely used for various biomedical applications.

Phenolic acids have been considered in recent years as potentially safe and effective cross-linkers of biopolymers. Tannic acid (tannic acid, tannin) is a naturally sourced compound constructed of a glycoside of gallic acid and glucose ring. It has bioactive properties such as antibacterial and antiviral [11]. It was reported that tannic acid improves the regeneration of the skin, for instance, in burn treatment [12]. Moreover, it was considered an effective cross-linker of biopolymers such as chitosan [13], collagen [14], hyaluronic acid [15], starch [16], and silk fibroin [17]. Tannic acid was selected by us to study the effectiveness of its use as a cross-linker for sodium alginate to improve the materials’ properties. 

The aim of the study was to obtain alginate-based films enriched by tannic acid as a cross-linker. The hypothesis was that the addition of tannic acid would improve the physicochemical properties and the biocompatibility of materials compared to pure alginate-based films. Materials were characterized by microscopic observation and roughness calculation. Moreover, the thermal properties and the total tannic acid release were determined. The biocompatibility was studied in material contact with blood and the observation of platelet adhesion to the film surface. It is important to study the physicochemical properties and the blood compatibility to consider the safety of sodium alginate/tannic acid-based films for biomedical application. 

## 2. Materials and Methods

### 2.1. Materials

Alginic acid sodium salt (SA; marine sourced from algae) and tannic acid (TA) were purchased from Sigma-Aldrich (Hamburg, Germany). Glycerin, diiodomethane, and ethanol were purchased from POCh (Poznań, Poland). Folin–Ciocalteu and sodium carbonate were purchased from Sigma-Aldrich (Germany). 

### 2.2. Samples Preparation

Sodium alginate and tannic acid were dissolved in 0.1 M acetic acid. Sodium alginate and tannic acid were mixed in the weight ratios of 90:10, 80:20, and 70:30. Solutions were mixed for 2 h on the magnetic stirrer. The mixtures were split on the plastic holders for solvent evaporation in room conditions. Thin films were obtained with the thickness of 0.13 ± 0.01 mm. 

### 2.3. Scanning Electron Microscopy (SEM)

A scanning electron microscope (SEM; LEO Electron Microscopy Ltd., Cambridge, UK) was used to observe the surface morphology and the cross-section of all the obtained films. SEM was also used to observe the platelet adhered to the material surface. In both analyses, films were sputter-coated with gold prior to the observation.

### 2.4. Atomic Force Microscopy (AFM)

The topography of sodium alginate/tannic acid films was observed by using the scanning probe microscope equipped with a Nanoscope IIIa controller (Version 1.40, Digital Instruments, Santa Barbara, CA, USA) operating in the tapping mode (room conditions; a scan rate of 1.97 Hz; a spring constant 2–10 N/m of silicon tips). The roughness of the surface of the film was calculated from 1 μm × 1 μm scanned area using Nanoscope software.

### 2.5. Differential Scanning Calorimeter (DSC)

Differential scanning calorimetry measurements were carried out with differential scanning calorimeter equipment (NETZSCH Phoenix DSC 204 F1, Selb, Germany) at the heating rate of 10 °C/min, temperature range from 20 °C to 250 °C in a nitrogen atmosphere with the flow of 40 mL/min. The samples (n = 5, weight 1.0–1.5 mg) were placed in the aluminum measuring pans.

### 2.6. Total Tannic Acid Release 

The tannic acid concentration was determined by the Folin–Ciocalteu method with the use of a standard curve. To study the tannic acid release, films were immersed in simulated body fluid (SBF; pH = 7.4). To perform this analysis, Folin–Ciocalteu (0.5 mL) reagent was mixed with Na_2_CO_3_ (1 mL), sample (1 mL), and distilled water to complete 10 mL. The mixture was then stored in 40 °C for 30 min. The concentration of tannic acid released was measured by UV–Vis spectrophotometer (UV-1800, Shimadzu, Reinach, Switzerland) at 725 nm with the use of a standard curve. 

### 2.7. Hemolysis 

Anticoagulated whole sheep blood was used to determine the effect of materials on red blood cell hemolysis. Into the test tube containing 0.2 mL of blood and 10 mL of saline salt, materials (10 mm × 10 mm) were added and left for contact time for 60 min in 37 °C (OD_specimen_). Tubes with the same amount of blood and saline salt without materials were left as negative control (OD_negative_). Positive control (OD_positive_) consisted of 0.2 mL of blood and 10 mL of distilled water. After incubation, test mixtures were centrifuged at 10,000 rpm for 10 min. The absorbance of the supernatants was measured by using the microplate reader Multiscan FC (Thermo Fisher Scientific, Waltham, MA, USA) at 540 nm. Each sample was prepared in triplicate. Then, the hemolysis rate was calculated using this equation:$$\mathrm{Rate}\;\mathrm{of}\;\mathrm{hemolysis}\;\left[\%\right]=\frac{{\mathrm{OD}}_{\mathrm{specimen}}-{\mathrm{OD}}_{\mathrm{negative}}}{{\mathrm{OD}}_{\mathrm{positive}}-{\mathrm{OD}}_{\mathrm{negative}}}\times 100$$

### 2.8. Platelet Adhesion Test

The whole blood was centrifuged at 2500 rpm for 10 min to obtain platelet-rich plasma (PRP). The samples with dimensions of 7 mm × 7 mm × 2 mm were sterilized using absolute ethanol. Samples were then dried for 5 min, placed in 12 well plates, and equilibrated for 6 h in 3 mL phosphate-buffered saline (PBS) at room temperature. Then, PBS was replaced with 3 mL of PRP. Samples were incubated for 90 min at 37 °C. After that, PRP was removed, and samples were washed with PBS (3 mL). Obtained cells were fixed using 2.5% glutaraldehyde for 30 min [18]. After fixing, the samples were washed with PBS and subsequently dehydrated using gradient washing in ethanol-water solutions. 

## 3. Results

### 3.1. Scanning Electron Microscope (SEM)

The surfaces (Figure 1) of each film were homogeneous and flat with uniform distribution of sodium alginate and tannic acid. Both components can be considered as miscible, as no precipitation was observed on the surface of the films or in their cross-sections (Figure 2). Changing the tannic acid content did not influence the films’ homogeneity.

### 3.2. Atomic Force Microscopy (AFM)

The highest roughness was noticed for the film based on pure sodium alginate (Table 1). The addition of tannic acid smoothed the surface, displayed by lower roughness parameters compared to 100SA (Figure 3). Moreover, higher tannic acid resulted in slight increases in Ra and Rq. 

### 3.3. Differential Scanning Calorimetry (DSC)

Thermal properties of the material are important to consider its stability. For each type of film, two peaks were observed on the DSC curve. The highest maximum temperature of the first stage process was noticed for sodium alginate-based films (Table 2). The addition of tannic acid resulted in the decrease of T_1_. The dependence on the material composition and the presence of tannic acid on the enthalpy were not observed. The presence of the first peak was related to the evaporation of water molecules bound to the material [19]. The second stage transformation was related to the degradation process of the material. The addition of tannic acid increased the maximum temperature of the second peak. Additionally, the enthalpy of the second process was changed, and it decreased with the increasing content of tannic acid. It was caused by the lower mass fraction of sodium alginate in the film sample with an increase in tannic acid content.

### 3.4. Total Tannic Acid Release

The tannic acid concentration released from the material depended on the initial content of phenolic acid in the material composition (Figure 4). It increased proportionally to its presence in the material. The highest concentration was noticed for the material composed of 70SA/30TA. In the initial time, the release was similar for each type of film containing tannic acid; however, after 24 h, the differences increased. The release percentage of tannic acid from the material composed of 90SA/10TA was initially 2% and, in the end of the experiment, it was around 10%. The release percentage for 80SA/20TA composition was initially around 4%; after 120 h, it was 20%. For the 70SA/30TA mixture, the release percentage was initially 5% and, in the end of the experiment, it was around 30%. It can be noticed that the release percentage was proportional to the initial content of tannic acid in the material.

### 3.5. Hemolysis 

The lowest hemolysis was noticed for the sodium alginate-based films (Table 3). The addition of tannic acid increased the hemolysis. This indicates that high tannic acid content in the material may result in the erythrocytes lysis in the range higher than 5%. However, all films proposed by us showed hemolysis lower than 5% and thereby can be considered for biomedical application [20].

### 3.6. Platelet Adhesion Tests

SEM images allowed us to observe the platelet adhesion on the film surface. The adhesion was noticed for films based on sodium alginate (Figure 5). However, the addition of tannic acid significantly decreased the platelet adhesion. This suggests that materials which contain phenolic acid are safer for biomedical application than materials based on pure sodium alginate. 

## 4. Discussion

Mixing different components should result in the obtainment of homogeneous materials. It is essential to consider the preparation method of a material as effective. Scanning electron microscope observation allowed to observe the morphology of the film by its surface and study of its cross-section. Sodium alginate/tannic acid-based films were considered as homogeneous, as we did not notice any inequalities. Moreover, the material had a compact structure in the cross-section. Generally, it was previously noticed that tannic acid might be mixed with other polysaccharides, e.g., chitosan [21], hyaluronic acid [22], and cellulose [23]. 

Atomic force microscopy may be used as an effective tool to study surface roughness that appears smooth to the naked eye. The surface of the material is critical, as it has direct contact with the surrounding. It was reported that cells increasingly adapt to larger surface roughness [24]. However, during the use of products, there is a competition between cells and bacteria for adhesion to the surface. The bacterial cells easily adhere to smooth surfaces. Tannic acid addition results in the decrease of surface roughness. It interacts with alginate and thereby changes the polymeric chain organization. Similar results were observed by us for the films composed of chitosan cross-linked by tannic acid, where the increasing amount of tannic acid in the material resulted in the decrease of both Ra and Rq parameters [25]. Thereby, it is important to carry out microbial studies to determine the antibacterial properties of sodium alginate/tannic acid films as well as the biofilm formation on the materials’ surface. 

Two step degradation during the heating is characteristic for polysaccharides. The first step is related to the loss of water molecules bonded to the material [26]. The second step is related to the changes in polymeric chains which depend on the material composition. Tannic acid acts as a sodium alginate cross-linker. The improvement of thermal properties was observed for the SA/TA mixture compared to pure SA. 

Tannic acid interacts with sodium alginate by the strong hydrogen bonds formation. However, it is released from the material in aqueous conditions. Similar observations were made by us for chitosan/tannic acid films where the released concentration of phenolic acid increased with increasing immersion time [27]. This is the advantage of the proposed materials, as tannic acid may act as a bioactive compound and influence the surrounding tissues and cells. 

Hemolysis rate is a crucial parameter to consider regarding material safety in biomedical applications [28]. The low hemolysis rate of the material suggests that it causes limited damage to red blood cells and has better blood compatibility [29]. All materials tested by us showed a hemolysis rate below 5% and can be considered as safe for biomedical application. The platelet adhesion is the next important parameter. Protein adsorption is the first event that occurs when the surface of the material comes into contact with blood [30]. In the platelet adhesion studies, we observed that it was reduced by the addition of tannic acid to sodium alginate. However, a higher hemolysis rate may indicate that the presence of tannic acid inhibited the adhesion and, at the same time, caused the destruction of erythrocytes. Similar results were obtained by Yang et al. [31]. Authors underlined that the reduction of platelet adhesion on the material surface is of tremendous scientific interest in the field of blood-contacting biomaterials. Tannic acid presence resulted in the reduction of platelet adhesion. Observations are in line with our research. The hydrogen bonds formed between alginate and tannic acid limited the number of platelet-binding sites, and thereby the adhesion was reduced [32].

In this study, we confirmed the safety of the material used in a short-time application (less than 90 min). To consider its application for a longer period of time, it is necessary to study the safety during the blood contact test as the tannic acid concentration released from the material is changed and may cause toxicity. 

## 5. Conclusions

Sodium alginate-based films may be modified by tannic acid addition. Obtained films had a homogeneous structure. Moroever, the addition of tannic acid to the sodium alginate improved the thermal properties of films. The release of tannic acid from the material was noticed; however, it did not cause toxicity in contact with blood. The important conclusion is that modification of sodium alginate such as tannic acid addition as a cross-linker is safe and does not change the consideration of the material as being appropriate for biomedical application. 

## Figures and Tables

**Figure 1 materials-14-04905-f001:**
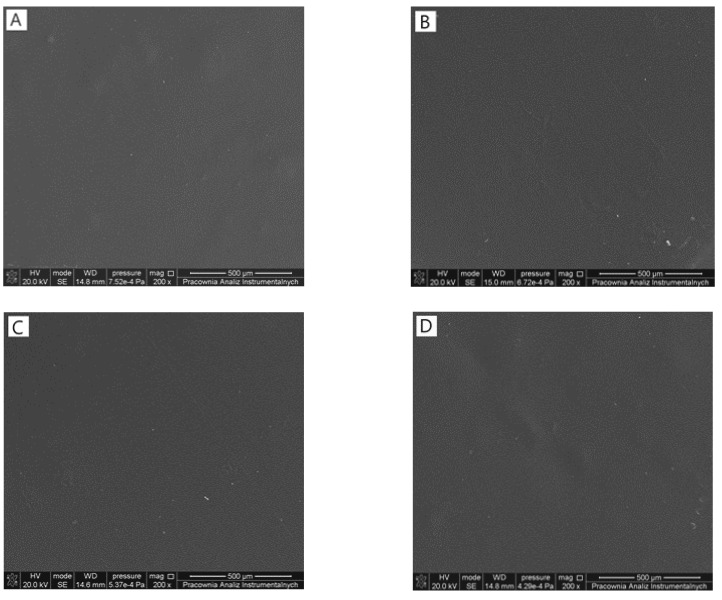
SEM images of (**A**) 100SA (**B**) 90SA/10TA (**C**)80SA/20TA (**D**) 70SA/30TA.

**Figure 2 materials-14-04905-f002:**
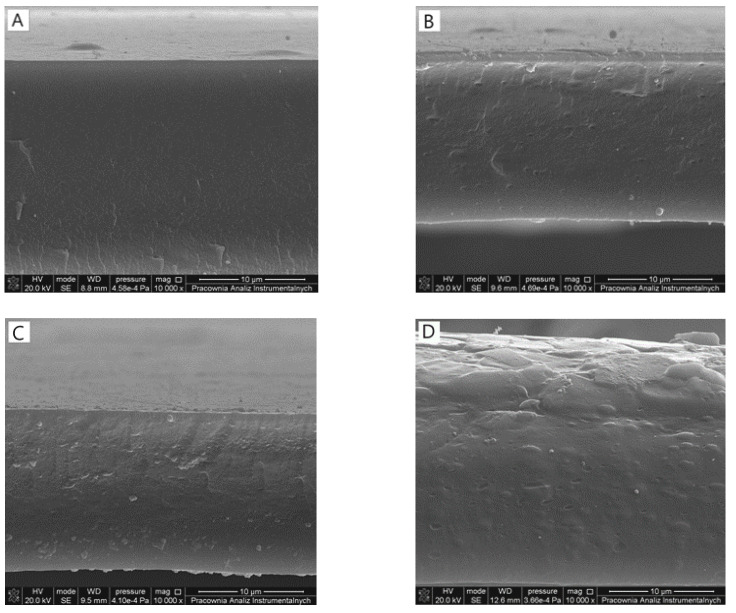
SEM images of cross-section of films composed of (**A**) 100SA (**B**) 90SA/10TA (**C**) 80SA/20TA (**D**) 70SA/30TA.

**Figure 3 materials-14-04905-f003:**
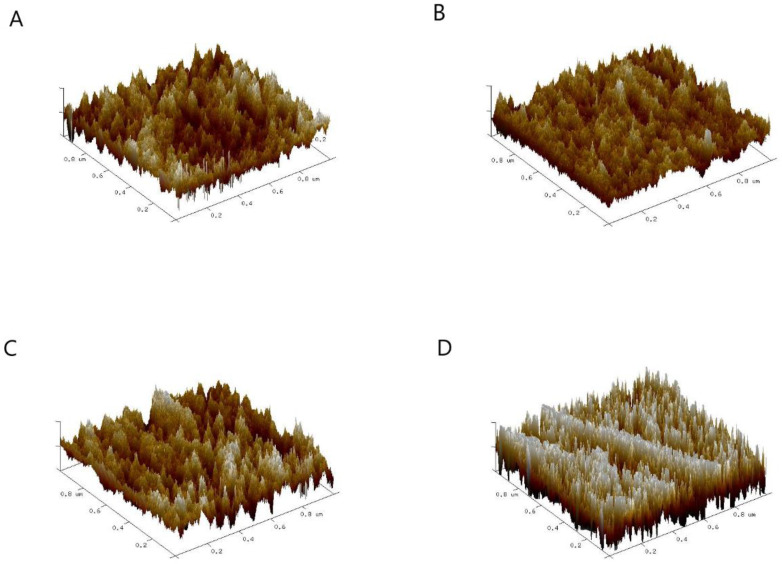
AFM three-dimensional images (1 µm × 1 µm) of (**A**) 100SA (**B**) 90SA/10TA (**C**) 80SA/20TA (**D**) 70SA/30TA (the presented images are representative for five specimens).

**Figure 4 materials-14-04905-f004:**
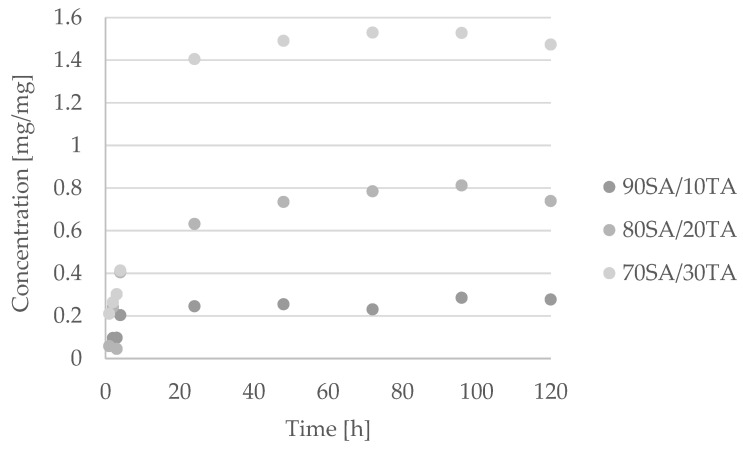
Total polyphenols released from sodium alginate/tannic acid films.

**Figure 5 materials-14-04905-f005:**
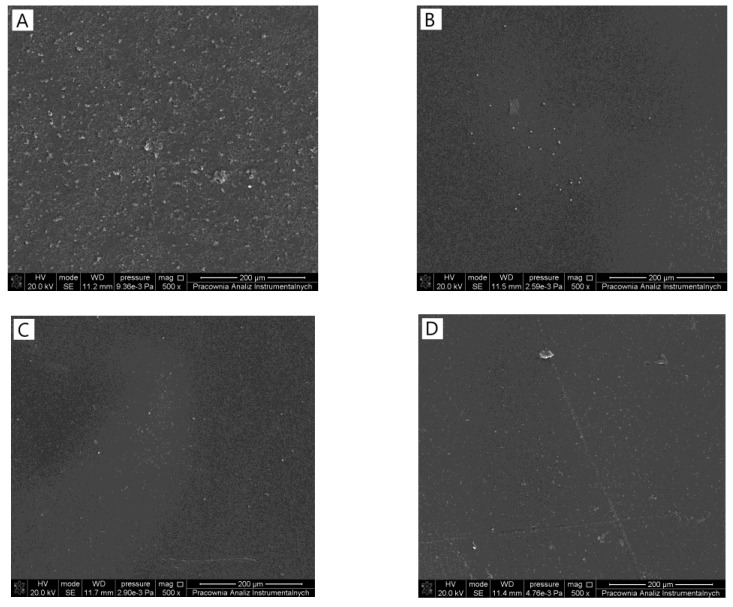
The morphology of the platelets adhered to the surface of (**A**) 100SA (**B**) 90SA/10TA (**C**) 80SA/20TA (**D**) 70SA/30TA (magnification: 1000×).

**Table 1 materials-14-04905-t001:** Roughness parameters (Ra and Rq) of sodium alginate/tannic acid films (n = 5; * significantly different from 100SA—*p* < 0.05).

Specimen	Ra (nm)	Rq (nm)
100SA	22.80 ± 0.07	28.30 ± 0.11
90SA/10TA	4.55 ± 0.02 *	5.62 ± 0.03 *
80SA/20TA	4.75 ± 0.02 *	6.06 ± 0.02 *
70SA/30TA	4.93 ± 0.03 *	6.31 ± 0.02 *

**Table 2 materials-14-04905-t002:** The maximum temperature of the process (T) and the enthalpy of the processes (ΔH) measured during the samples heating by differential scanning calorimetry.

Specimen	T_1_ (°C)	ΔH_1_ (mW/mg)	T_2_ (°C)	ΔH_2_ (mW/mg)
100SA	77.9	1.342	196.0	0.3357
90SA/10TA	68.6	1.238	204.7	0.2747
80SA/20TA	71.2	0.993	200.8	0.2188
70SA/30TA	68.3	1.050	200.4	0.1376

**Table 3 materials-14-04905-t003:** Hemolysis rate for each type of film in contact with blood films (n = 5; * significantly different from 100SA—*p* < 0.05).

Specimen	Hemolysis Rate (%)
100SA	0.66 ± 0.12
90SA/10TA	1.12 ± 0.09
80SA/20TA	1.69 ± 0.21 *
70SA/30TA	2.41 ± 0.19 *

## Data Availability

The data presented in this study are available on request from the corresponding author. The data are not publicly available due to project realization.

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
