# Peer review of "The Study of Physicochemical Properties and Blood Compatibility of Sodium Alginate-Based Materials via Tannic Acid Addition"

_materials, 2021, doi:10.3390/ma14174905_

Round 1

Reviewer 1 Report

In the paper, the authors modified the alginate film with the addition of tannic acid and characterize the films using SEM, AFM, DSC, Folin-Ciocalteu method for release profile, hemolysis rate and platelet adhesion. Overall, the results presented in the paper lack significance and conclusions were drawn prematurely. Additional experimental work and more elaborated discussions are needed to be considered adequate for this Journal. Specific issues are listed below:

  1. SEM figure 1 and 2 were of low quality. It was not possible to conclude the films were homogenous.
  2. The decrease of the enthalpy of the second peak does not mean the thermal property were improved. It is merely caused by the lower mass fraction of sodium alginate in the sample with the increase in tannic acid content.
  3. As indicated by the tannic acid release profile, the initial burst release were similar for all three films while over long times (10-120 hours), higher tannic acid samples have higher released amounts. However, hemolysis and platelet adhesion were all only tested after a short exposure. (<90 mins). This is not sufficient to evaluate the negative effects of tannic acid and cannot conclude that the proposed films are biocompatible.

Author Response

1) SEM figure 1 and 2 were of low quality. It was not possible to conclude the films were homogenous.

Thank you for the comment. We improved the image quality.

2) The decrease of the enthalpy of the second peak does not mean the thermal property were improved. It is merely caused by the lower mass fraction of sodium alginate in the sample with the increase in tannic acid content.

We highly appreciate your comment. It is now corrected.

3) As indicated by the tannic acid release profile, the initial burst release were similar for all three films while over long times (10-120 hours), higher tannic acid samples have higher released amounts. However, hemolysis and platelet adhesion were all only tested after a short exposure. (<90 mins). This is not sufficient to evaluate the negative effects of tannic acid and cannot conclude that the proposed films are biocompatible.

Thank you very much for the comment. We agree. The time of the experiment was selected by us due to the potential application of obtained material as a wound dressing. The time of contact with blood we estimated for 1 or 1.5h. It is now discussed.

Reviewer 2 Report

The authors have attempted to investigate how tannic acid can improve the properties of sodium alginate-based material. Different studies have been conducted to prove that tannic acid helps to improve the properties of sodium alginate.

However, the question is why tannic acid and not any other biodegradable synthetic polymer or biopolymer? It is essential to lay that background in introduction section, to provide readers with comprehensive understanding of problem and proposed alternative or solution.

AFM study: Authors have proved that film based on sodium alginate with tannic acid has lower roughness than that of without tannic acid. However, discussion says that cells adapt better to rough surface than smooth. Then why do you need smooth surfaced material?  Besides, bacterial cells easily adhere to smooth surface than rough. Please discuss in detail.

Release study: Authors have mentioned that tannic acid interacts with sodium alginate with hydrogen bond, and it is released in aqueous solution. Release of phenolic compound increases with immersion time. So, the question is what is the advantage here? Please discuss in detail.

Authors should consider rewriting manuscript with proficient discussion using appropriate references. I would highly recommend reading recent articles on the use of tannic acid polymer composition for biomedical applications.

There are many typographical and grammatical errors as well, please rectify them.

Some examples;

Line 63: Please mention the thickness of the thin film.

Line 95: replace ‘od’ with ‘of’.

Lines 143, 163, 171 and so on.

Author Response

1) However, the question is why tannic acid and not any other biodegradable synthetic polymer or biopolymer? It is essential to lay that background in introduction section, to provide readers with comprehensive understanding of problem and proposed alternative or solution.

Thank you very much for the suggestion. We added the paragraph about tannic acid. We hope it is now acceptable.

2) AFM study: Authors have proved that film based on sodium alginate with tannic acid has lower roughness than that of without tannic acid. However, discussion says that cells adapt better to rough surface than smooth. Then why do you need smooth surfaced material?  Besides, bacterial cells easily adhere to smooth surface than rough. Please discuss in detail.

Thank you very much for the comment. We discussed that based on the recent studies it is known that roughness may determine the attachment of human cells or bacteria. We obtained surface smoother when alginate was mixed with tannic acid compared to the pure alginate. However, the surface was not ideally smooth. We assumed that it is important to carry out the microbial studies to determine if materials show antibacterial activity and if the biofilm is formed on their surface. We hope it is now acceptable.

3) Release study: Authors have mentioned that tannic acid interacts with sodium alginate with hydrogen bond, and it is released in aqueous solution. Release of phenolic compound increases with immersion time. So, the question is what is the advantage here? Please discuss in detail.

In our opinion, it is an advantage as tannic acid is a bioactive compound. When it is released from the material it has a possibility to influence the surrounding. We expect that it has antimicrobial activity, however, we did not carry our microbial tests in the paper. It is now written in the paper.

4) Authors should consider rewriting manuscript with proficient discussion using appropriate references. I would highly recommend reading recent articles on the use of tannic acid polymer composition for biomedical applications.

Thank you very much for the comment. We agree that new references would be valuable. Thereby, we added new references to the manuscript.

5) Line 63: Please mention the thickness of the thin film.

Thank you for the comment. It is now corrected.

6) Line 95: replace ‘od’ with ‘of’.

We apologize for the mistake. It is now corrected.

7) Lines 143, 163, 171 and so on.

Thank you for the comment. The manuscript was revised carefully and corrected.

Reviewer 3 Report

The research activity - focused on the preparation and characterization of alginate-based materials with tannic acid - is interesting and fits well with the scopes of the journal. However, the text requires minor revisions as concerning both data and their discussion.

In section 1-Introduction a brief introduction of tannic acid and the reason why it has been selected for this study should be reported.

In Section 4-Discussion the results should be discussed deeper and with more accuracy, for example the AFM data are not commented but it is only reported the raw data and some generic observations from literature. As well as the platelet adhesion studies.

Few detailed comments are reported hereafter:

  • Pg 2 line 63 Why is the thickness of sample reported as xx?
  • Pg 4 line 122 Specify that micrographs are related to crossection views
  • Pg 4 line 126 Specify where precipitation was not observed, is it in crossection views?
  • Pg 6 line 166 It is should be interesting to report the release percentage

Author Response

1) In section 1-Introduction a brief introduction of tannic acid and the reason why it has been selected for this study should be reported.

Thank you for the comment. We added a separate paragraph in the Introduction section.

2) In Section 4-Discussion the results should be discussed deeper and with more accuracy, for example the AFM data are not commented but it is only reported the raw data and some generic observations from literature. As well as the platelet adhesion studies.

Thank you very much for the comment. We improved the discussion in the discussion part of the manuscript.

3) Pg 2 line 63 Why is the thickness of sample reported as xx?

We apologize for the mistake. It is now corrected: 0.13 ± 0.01 mm.

4) Pg 4 line 122 Specify that micrographs are related to crossection views

Thank you for the suggestion. It is corrected.

5) Pg 4 line 126 Specify where precipitation was not observed, is it in crossection views?

Thank you for the comment. We specified: Both components can be considered as miscible as precipitation on the film’s surface as well as in their cross-section was not observed.

6) Pg 6 line 166 It is should be interesting to report the release percentage

Thank you very much for the comment. We discussed the release percentage.

Reviewer 4 Report

Dear authors,
The manuscript related to improving the properties of sodium alginate-based materials via tannic acid addition has been reviewed. As one of the selected reviewers, I read your manuscript carefully. While your research has addressed an important subject, I appreciate the authors' performing research that is fruitful for academia and industry. Some clarifications and improvements would make the manuscript considerable for publications.
The title is too general; please consider mentioning what properties or try to make it more specific.
The authors should address the main issue and describe the need for this research in the introduction.
The introduction has not been written critically. The introduction rephrases the work that has been done. I could not find any discussion or opinion about previous research.
The most serious flaw is the lack of a specific research problem compare to previous research. In the absence of such a research problem, it is unclear what new problems you intended to solve and the dimensions of these problems.
Consider citing Figures while discussing the results so that the reader may easily follow you.
Add some marks to Figure 1 and highlight the difference between these four images, rather than pasting original images. The original raw images may be added as supplementary or data with the paper.
Please correct, "Thin films were obtained with the thickness xx mm."
What fiber was brought into consideration? Kindly add the name and source of material in the Materials and Methods section.
I could not find any comparison of the proposed model with the existing technology. Can the author propose a comparison table or schematics to gain the readers' attention?
Add concrete conclusions.
Add statistical error in Table 1.
What are the prospects? Add a separate section or briefly add few sentences in the conclusion section? Also, no research limitation is explained aligned with the research problem. The research limitations describe what dimensions of the problem are excluded by you and your study's boundaries.
I look forward to receiving the revision and reviewing the newer version.
Best of luck

Author Response

1) The title is too general; please consider mentioning what properties or try to make it more specific.

Thank you for the comment. Our proposition for the change is: The study of physicochemical properties and blood compatibility of sodium alginate-based materials via tannic acid addition

2) The authors should address the main issue and describe the need for this research in the introduction.

Thank you for the comment. It is now discussed. 

3) The introduction has not been written critically. The introduction rephrases the work that has been done. I could not find any discussion or opinion about previous research.

Thank you very much for the comment. We added part related to the tannic acid studies. We hope it is acceptable now.

4) The most serious flaw is the lack of a specific research problem compare to previous research. In the absence of such a research problem, it is unclear what new problems you intended to solve and the dimensions of these problems.

We underlined the hypothesis of the study and hope that now it is more clear for the readers.

5) Consider citing Figures while discussing the results so that the reader may easily follow you.

Thank you very much for the suggestion. It is now corrected.

6) Add some marks to Figure 1 and highlight the difference between these four images, rather than pasting original images. The original raw images may be added as supplementary or data with the paper.

Thank you very much for the comment. In our opinion the consideration of material homogeneity is important factor. Thereby, we placed the SEM images. We did not notice any differences in the images A-D what provides the information that higher tannic acid addition do not change the material homogeneity. It is now discussed.

7) Please correct, "Thin films were obtained with the thickness xx mm."

We apologize for the mistake. It is now corrected: 0.13 ± 0.01 mm.

8) What fiber was brought into consideration? Kindly add the name and source of material in the Materials and Methods section.

For the films preparation we used tannic acid and sodium alginate in the powder form. Both components were purchased from Sigma-Aldrich company. It is written in the “Materials” section.

9) I could not find any comparison of the proposed model with the existing technology. Can the author propose a comparison table or schematics to gain the readers' attention?

Thank you very much for your comment. We discussed the recent studies related to the phenolic acids use as biopolymers cross-linkers. We also wrote the hypothesis of our studies in the present version of the manuscript. We hope it is acceptable and clear for readers.

10) Add concrete conclusions.

Thank you very much for the comment. Conclusions are now added.

11) Add statistical error in Table 1.

Thank you for the comment. The statistical error is now written.

12) What are the prospects? Add a separate section or briefly add few sentences in the conclusion section? Also, no research limitation is explained aligned with the research problem. The research limitations describe what dimensions of the problem are excluded by you and your study's boundaries.

The main boundary of the material application is that we studied the short-time exposure on the blood contact. It is probable that longer time of application would indicate the toxicity. Thereby, it has to be studied. We confirmed that the material is safe to be used for the contact time below 90 min. We added this aspect in the manuscript.

Round 2

Reviewer 1 Report

The authors have addressed all of my concerns raised in the review of the first version.

Author Response

Thank you very much

Reviewer 2 Report

The authors have addressed all the concerns very well. However while addressing comment 3, authors have mentioned that tannic acid has possible influence on surrounding (line 251-52). What do authors mean by surrounding? a brief explanation is necessary (please elaborate continuing line 252).

"In our opinion, it is an advantage as tannic acid is a bioactive compound. When it is released from the material it has a possibility to influence the surrounding. We expect that it has antimicrobial activity, however, we did not carry our microbial tests in the paper. It is now written in the paper".

Author Response

The authors have addressed all the concerns very well. However while addressing comment 3, authors have mentioned that tannic acid has possible influence on surrounding (line 251-52). What do authors mean by surrounding? a brief explanation is necessary (please elaborate continuing line 252).

"In our opinion, it is an advantage as tannic acid is a bioactive compound. When it is released from the material it has a possibility to influence the surrounding. We expect that it has antimicrobial activity, however, we did not carry our microbial tests in the paper. It is now written in the paper".

Thank you very much for the comment. We made corrections:

"It is the advantage of proposed materials as tannic acid may act as a bioactive compound and influence the surrounding tissues and cells."

We hope it is acceptable now.

Reviewer 4 Report

The authors have improved the manuscript as per the reviewer's guidance; therefore, the manuscript might be considered for publication after a minor spell check.

Moreover, the self-citations were increased in the revised version which must be minimized.

Author Response

Thank you very much for the comment. We made the grammar corrections and also minimized the self-citations.